# Generalized Deep Neural Network Model for Cuffless Blood Pressure Estimation with Photoplethysmogram Signal Only

**DOI:** 10.3390/s20195668

**Published:** 2020-10-04

**Authors:** Yan-Cheng Hsu, Yung-Hui Li, Ching-Chun Chang, Latifa Nabila Harfiya

**Affiliations:** 1Department of Electrical Engineering, National Central University, Taoyuan 32001, Taiwan; bill.yancheng.hsu@gmail.com; 2Department of Computer Science and Information Engineering, National Central University, Taoyuan 32001, Taiwan; yunghui@csie.ncu.edu.tw; 3Department of Electronic Engineering, Tsing Hua University, Beijing 100084, China; c.c.chang.phd@gmail.com

**Keywords:** photoplethysmogram (PPG), cuffless blood pressure (BP) estimation, cardiovascular disease (CVD) prevention, artificial neural network, wearable biomedical applications

## Abstract

Due to the growing public awareness of cardiovascular disease (CVD), blood pressure (BP) estimation models have been developed based on physiological parameters extracted from both electrocardiograms (ECGs) and photoplethysmograms (PPGs). Still, in order to enhance the usability as well as reduce the sensor cost, researchers endeavor to establish a generalized BP estimation model using only PPG signals. In this paper, we propose a deep neural network model capable of extracting 32 features exclusively from PPG signals for BP estimation. The effectiveness and accuracy of our proposed model was evaluated by the root mean square error (RMSE), mean absolute error (MAE), the Association for the Advancement of Medical Instrumentation (AAMI) standard and the British Hypertension Society (BHS) standard. Experimental results showed that the RMSEs in systolic blood pressure (SBP) and diastolic blood pressure (DBP) are 4.643 mmHg and 3.307 mmHg, respectively, across 9000 subjects, with 80.63% of absolute errors among estimated SBP records lower than 5 mmHg and 90.19% of absolute errors among estimated DBP records lower than 5 mmHg. We demonstrated that our proposed model has remarkably high accuracy on the largest BP database found in the literature, which shows its effectiveness compared to some prior works.

## 1. Introduction

According to a statistical report from the World Health Organization (WHO), cardiovascular disease (CVD) is the leading cause of death worldwide, with an estimated 17.9 million people dying from CVD in 2016, representing 31% of global deaths [1]. Early detection and treatment could effectively reduce the incidence and mortality rates. As a result, there is an urgent need for efficient and reliable means of managing cardiovascular risk factors, such as diabetes, hypertension or hyperlipidemia.

Blood pressure (BP) is considered to be one of the most important contributory risk factors and, therefore, real-time monitoring of BP plays a crucial role in saving people from premature death caused by CVD. The most common automated BP measurement devices are cuff based, as shown in Figure 1. They take about one to two minutes to produce one set of diastolic blood pressure (DBP) and systolic blood pressure (SBP) measurements before making another measurement. This type of measurement can be time-consuming and is often inaccurate [2]. In view of these issues, some neural network-based regression models were developed and they shortened the time interval of BP measurement from 1–2 min to less than 10 s. The accuracy of these regression models also met the criteria (i.e., a protocol of requirements for the evaluation of BP measuring devices) defined and recommended by the British Hypertension Society (BHS) [3,4]. While these models produced satisfactory performance in terms of operation time and accuracy for real-time BP estimation, they may not be practical solutions. The key reason is that these models are required to estimate physiological parameters from electrocardiogram (ECG) and photoplethysmogram (PPG) signals, implying that physiological parameter extraction from two different sensors is needed, and this solution incurs substantial cost.

The other reason is from the basic theory of the classical method of extracting physiological parameters from ECGs and PPGs, which relies heavily upon the theory based on pulse wave velocity (PWV). PWV is the velocity of pressure pulse initiated by the heartbeat, propagating through arteries, similar to a pipe with elastic walls. PWV has been proved to be highly related to BP and their correlation can be represented as [5,6]:(1)$$\mathrm{PWV}=\sqrt{\frac{E\cdot h}{2\cdot r\cdot \rho }},$$
where r, h, E and ρ denote the radius of the artery, the thickness of the artery, the elastic modulus of the arterial wall and the density of blood in of the artery, respectively. There are several existing approaches that can calculate PWV and, among them, the most widely used one for PWV calculation is pulse wave transit time, commonly referred to as pulse transit time (PTT). The relation between PWV and PTT can be represented as follows [7]:(2)$$\mathrm{PWV}=\;\frac{d}{\mathrm{PTT}}\;,$$
where PTT is the time interval between a pulse wave being detected by two sensors and d is the distance between the sensors on the artery. In (1), the elastic modulus E is assumed as a constant when in fact the value of E in the artery is testified to be exponentially escalated with the blood pressure, as follows [8]: (3)$$E\left(P\right)=E_{0}\cdot e^{\alpha \cdot P},$$
where $E_{0}$ denotes the elastic modulus at 0 mmHg (the unit of blood pressure) and α is a parameter larger than zero that is closely related to arterial stiffness. The stiffer the artery, the greater the value of α. We can find a nonlinear relationship between blood pressure P and PTT after we substitute (1) and (2) into (3) to be
(4)$$P=-\frac{2}{\alpha }\cdot \ln\left(\mathrm{PTT}\right)+\frac{1}{\alpha }\cdot ln\left(\frac{2\cdot r\cdot \rho }{E_{0}\cdot h}\cdot D^{2}\right),$$

Though it may seem easy from the theoretical perspective, it would be inconvenient and almost impossible to use (4) directly since there is no way to get all the person-dependent variables in a short period of time. The other approach is done by extracting a set of representative time indices, including PTT (p), PTT (d) and PTT (f), as shown in Figure 2, from the relative location between the PPG and ECG signals [9,10]. However, it is still a very challenging task since the ECG waveform, in particular, has higher variability [7] and its accuracy is still limited for clinical uses [11].

Notwithstanding that devices such as, but not limited to, wearable devices that have the capability of recording both ECG and PPG signals are now thriving in the market, these are still highly priced and prevent users from having direct access to the data [12]. On the other hand, PPG sensors have been largely applied in wearable devices. They are popular as a low-cost but robust technology with full accessibility. Prior works have emphasized the relevance of a more detailed study of PPG signals only [13,14,15,16,17,18]. Hence, a ECG-free BP estimation model is preferable to improve the usability and to reduce the cost since the devices will no longer need additional biosensors for detecting ECG signals.

## 2. Literature Review

PPG is a non-invasive technique for measuring changes in blood volume due to the blood pulsatile nature of microvascular tissue under the skin [19]. The characteristics of the PPG waveform, along with its derivatives, have been discussed in [16,20,21]. Here, we can conclude that taking its first and second derivatives significantly helps in detecting the informative features in the PPG waveform. From a biomedical application perspective, [13] shows that PPG is an effective technique for diagnosing several CVDs and is able to be utilized in new medical tools such as the Internet of Things and biosensors. The clinical applicability of PPG is also verified in [18], which tried to distinguish individuals with congestive heart failure from healthy individuals by applying the concept of natural time analysis (NTA). NTA is applied to analyze a phase change or critical point in a complex system such as the human heart. The results obtained by PPG demonstrate a comparable value of accuracy to the results obtained by ECG. 

For a more specific BP estimation task, we can basically divide them into two approaches, feature-based and whole-based methods. In [14], five different features, which consist of the pulse area, pulse rising time, pulse width at 25% of pulse height, pulse width at 50% of pulse height and pulse width at 75% of pulse height, were extracted from a PPG segment. Machine learning methods, such as multiple linear regression (MLR), support vector machine (SVM) and regression tree, were then utilized for training and testing the data for estimating both DBP and SBP values, with the best overall accuracy being achieved using the regression tree. Another work, [22], uses several spectral and morphological features, such as systolic upstroke time and diastolic time. Using artificial neural network (ANN) architecture for fitting the features to simultaneously estimate the DBP and SBP, this method reduces the error from the other methods used as comparisons, such as linear regression and regression support vector machine (RSVM). On the contrary, the whole PPG waveform segment was extracted and used as the input of deep learning models in [15,17]. Both models comprised a convolutional neural network (CNN) and its modification to capture the spatial features of the waveforms. Both models achieved impressive accuracy, with a relatively low distribution of error as well.

## 3. Materials and Methods

The main flow of the proposed model is illustrated in Figure 3, which was composed of preprocessing, a feature extractor and a deep neural network predictor. Since the raw PPG might contain noise and long sequences, preprocessing is necessary for further feature extraction processes. In this section, a detailed explanation about each part is presented, with a summary introduced as follows:***Data preprocessing:*** This part comprises signal smoothing of raw PPG data and the removal of abnormal data following standard procedures suggested by [23]. Next, we partition the preprocessed PPG into an approximately 2.17 million heart cycles.***Feature extractor:*** Features from the preprocessed data are further extracted and selected as the input set.***Deep neural network predictor:*** We feed the feature set into a deep neural network predictor, which consists of five fully connected layers, and each layer contains 2000+ units of fully connected perceptrons, responsible for predicting BP in each heart cycle from 32 extracted physiological parameters.

### 3.1. Data Source

For every data-driven neural network application, the data themselves mean everything, affecting regression models from every perspective. Specifically, the diversity of data affects how generalized a model is and, the more variability the data has, the more generalized the model that can be trained. Based on this idea, the Multi-parameter Intelligent Monitoring in Intensive Care (MIMIC) II online waveform database, which has been refined and prescreened in the literature [9], and contains 12,000 data instances indicating unique subject records and an estimated more than 4 million heart cycles, was used in this study. Among this enormous dataset, we preselected 9000 data instances for training, validating and testing our deep neural network predictor.

### 3.2. PPG Raw Data Preprocessing

Prior to the actual process of estimating the blood pressure using PPG signal exclusively, we conduct a preprocessing to enhance the quality of the PPG. The raw PPG data preprocessing consists of four main steps, including noise removal, normalization, feature point detection, and partitioning. The detailed explanation of each step is presented as follows:
Noise removal: Fast Fourier transform (FFT) is applied to every PPG data segment to convert it from its time domain into the frequency domain. Let x[n], 0≤n≤N−1, represent the PPG, and the FFT of *x*[*n*] is denoted as X[k], 0≤k≤N−1. We remove the frequency components that are lower than 0 Hz or higher than 8 Hz by turning off those frequency components, as follows
(5)$$X_{r}\left[k\right]=\;\{\begin{array}{cc} X\left[k\right] & k\geq 8\\ 0 & otherwise \end{array}\;,$$ By removing this range of frequency, we aim at removing noise and the baseline wander. The PPG signal can then be restored into the time domain with inverse FFT (IFFT).Normalization and 1st and 2nd derivative of PPG calculation (denoted as “dPPG” and “sdPPG”): All the raw values of PPG are positive, so min–max normalization is applied to every PPG data segment. The equation of min–max normalization can be represented as (5):(6)$$x^{\prime }\;=\;(x\;-\;X_{m})/(X_{M}-\;X_{m}),$$
where *x* are data points in each PPG data segment {X} and $X_{m}$ and $X_{M}$ are the minimum and maximum values, respectively, in each PPG data segment {X}. After min–max normalization, the values of every PPG data segment are within the range [0 1] and dPPG and sdPPG (1st and 2nd derivative of waves of PPG) are calculated at the same time. Feature point detection: Before feature extraction, a few points should be marked and detected in every cycle of the heartbeat for every signal (PPG, dPPG and sdPPG) for cycle segmentation and alignment. Firstly, the systolic peaks of PPG waves of each heart cycle are marked by taking advantage of an algorithm mentioned in [24]. The correctness and validity of the systolic peak detection algorithm is of vital importance because the rest of the feature point detection algorithm is based on it. Secondly, the onset and offset valley points of PPG are detected by finding the minimum between two consecutive systolic peaks. Thirdly, with the valley points of PPG found, the location with the maximal and minimal slope values of PPG and dPPG can easily be derived by computing their gradients. Fourthly, the dicrotic notch points of PPG are detected by finding the secondary peaks of the sdPPG contour [20]. An example set of waveforms is shown in Figure 4.Partitioning and abnormal cycle removal: After feature points are located, each PPG data segment and its corresponding dPPG and sdPPG waves are partitioned into fragments by reserving each PPG data segment from one valley point of PPG to the next consecutive valley point of PPG. Abnormal heart cycles are also removed following the criteria mentioned in [23]. After abnormal cycle removal is done, the histograms of distribution of SBP and DBP are plotted, as seen in Figure 5, and approximately 2.17 million PPG, dPPG and sdPPG data fragments of heart cycles are obtained. 

### 3.3. Feature Extraction and Selection Index γ

#### 3.3.1. Feature Extraction

The candidate features are the 65 features proposed in past studies [10,16,25,26,27,28], which are reported to be highly related to blood pressure estimation [11]. Among them, we select 59 features, including *hr, t1, t2, t3, t4, t5, t6, t7, t8, AS, dAS, sdAS, DS, dDS, sdDS, S1, S2, AA, dAA, sdAA, DA, dDA, sdDA, RAAD, dRAAD, sdRAAD, PI, dPI, sdPI, dVI, sdVI, AID, dAID, sdAID, dDID, sdDID, PIR, dPIR, sdPIR, dRIPV, sdRIPV, AT, dAT, sdAT, DT, dDT, sdDT, dTVO, sdTVO, Slope_a, S3, S4, RtArea, NI, AI, AI1, RSD, RSC* and *RDC*. All the definitions of the 59 features are listed in Table 1. The extracted features are first standardized to value [−1 1] using Z-score normalization, as shown in (7), for each feature: (7)$$\;y^{\prime }\;=\;\frac{y-\;\mu }{\sigma },$$
where y are the elements in each feature and μ and σ are the mean value and standard deviation of each feature, respectively. Although the authors of [16] observed a phenomenon that the fluctuation in BP led to conspicuous changes in these 59 features, seemingly unveiling the close correlation between BP and these 59 features in their dataset, whether the same phenomenon will happen again in our experimental data source is still unclear and unpredictable due to the fact that the database we use is not only different but is also more diverse than the dataset mentioned in [16]. Consequently, statistical experiments on our dataset are recomputed and, moreover, an index γ is introduced to evaluate the degree of correlation between each feature and BP. 

#### 3.3.2. Selection Index γ

An index γ is introduced to ensure the validity of all selected features, and γ is defined as:(8)$$\;\gamma \;\equiv \;\frac{\sum_{i=\mu -\alpha }^{i=\mu +\alpha }\;\left\{\;\left|\;f\left(\;\frac{i}{\beta }\;\right)\;-\;\int_{\frac{i}{\beta }}^{\frac{i+10}{\beta }}ℊ\left(u\right)du\;\right|\right\}}{\sum_{x=\mu -\alpha }^{x=\mu +\alpha }1},$$
where:*f*(u) = probability mass function of standardized target feature, its estimated precision is down to k decimal places.ℊ(u) = probability density function of standard normal distribution.β = $10^{k+1}$. μ = 0, the mean of the standardized target feature.α = $10^{k}$Cσ , where σ is the standard deviation of the target feature (in the case of standardized features, μ and σ are equal to 0 and 1) and C is an integer. For evaluation, the definition of the values of features ranging from μ – Cσ to μ + Cσ is used.

Figure 5 shows the histogram distribution of SBP and DBP values in our dataset. From Figure 5, it seems that the distributions of SBP and DBP are close to the normal distribution and, as a result, σ is designed to check the degree of similarity between the standard normal distribution and the distributions of each standardized feature. The smaller the value of σ, the higher the degree of similarity. In fact, the basic concept of σ is to compute the mean absolute error (MAE) between the standard normal distribution and the distributions of each standardized feature within u = μ±Cσ. In our experiments, firstly, while computing σ, our standardized features are all computed to one decimal place. So, in our case, k = 1. In addition, since we plan to evaluate the similarity using a value within u = μ±3σ, in our case, C = 3. All the values of σ of the 59 features mentioned above are computed and listed in Table 1.

### 3.4. Deep Neural Network Predictor

After feature selection is done, the optimal feature set considered to be highly correlated to BP is obtained. The next step is to train a machine learning model which is able to predict the SBP and DBP values accurately given the selected features. In this work, we use a fully connected deep neural network regressor for this goal. As shown in Figure 3, our model is composed of multiple fully connected layers with activation function “ReLU”. Between input layers and output layers, each hidden layer contains 2048, 4096, 8192 and 2048 fully connected neurons, respectively.

#### 3.4.1. Introduction to Fully Connected Neural Network

As a matter of fact, a brain of a human comprises billions of neurons connected each other with synapses, and each neuron communicates through electrical currents. A special kind of machine learning model, called a neural network (NN), was proposed a long time ago to mimic the behaviors of neurons. A generic NN consists of perceptrons, mimicking the function of biological neurons, and an interconnected layered structure that connects every perceptron in one layer to another. Each perceptron contains a weighted vector W and a bias b, as seen in Figure 3, whose value gets updated iteratively during the training process. The correlation between the input and output of a fundamental perceptron can be formulated as (9):(9)$$O(I^{T})\;=\;\mathrm{act}(WI^{T}+\;b),$$
where $I^{T}$ is a transpose input vector of a perceptron, O($I^{T}$) is an output value of a perceptron, act() represents an activation function, and, in our case, activation function “ReLU” is applied in our deep neural network models. If every perceptron in one layer is connected to every perceptron in the next layer, such an NN is called a “fully connected neural network”, as seen in Figure 6. 

#### 3.4.2. Neural Network Selection

As the development of artificial intelligence has evolved, more and more different kinds of neural networks, such as fully connected networks, convolutional neural networks, and recurrent neural networks, have been proposed to tackle different kinds of problems. Among them, long short-term memory (LSTM) and fully connected neural networks are the most commonly applied regressors for building BP estimation models. Su et al. [29] constructed an LSTM-based model with high accuracy across 84 subjects by extracting classical PTT-related features mentioned in Section 1. On the other hand, Kurylyak et al. [30] used a fully connected neural network to build up a valid model across 15,000 cardiac cycles by utilizing temporal features extracted from PPG segments. The two different kinds of models actually have their pros and cons. In this paper, we decided to adopt the fully connected neural network as our regressor since it is easier to be implemented in wearable devices. The model structure is clean and easier to understand compared to LSTM, which enables software engineers to transfer and deploy the code to wearable devices. Another advantage of a fully connected network for BP estimation is that it takes inputs from only one cardiac cycle to estimate BP. On the other hand, the LSTM model usually takes inputs from several cardiac cycles before it outputs BP values, which causes a time delay when dealing with patients with a critical situation of CVD.

## 4. Experiments and Results

### 4.1. Feature Point Detection and Abnormal Cycle Removal

Following the methodologies mentioned in Section 3.3, one example of a result of a PPG and its corresponding dPPG and sdPPG marked with feature points is demonstrated in Figure 7 and the validity of the algorithm that helps us locate the dicrotic notch in every cardiac cycle by finding the corresponding secondary peaks of the sdPPG contour signal [20] is strengthened and verified by our statistical results after experimenting with normalized notch intensity across more than 2.17 million cycles in our dataset. The distribution of normalized notch intensity is shown in Figure 8. After feature point detection, partitioning and abnormal cycle removal are done, approximately 2,176,188 data fragments of the PPG, dPPG and sdPPG of a single cardiac heart cycle are obtained and the distributions of the corresponding SBP and DBP values are shown in Figure 5. One of the results of feature point detection is demonstrated in Figure 7. 

### 4.2. Characteristic Features of Cardiac Cycles

With the help of the σs computed and listed in Table 1, the process of feature selection becomes easier. The criterion for feature selection is to choose the features with the lowest σs. Indeed, 30 out of 32 selected features, including hr, AS, DS, AA, dAA, sdAA, DA, dDA, sdDA, PI, dPI, sdPI, dVI, sdVI, AID, dAID, sdAID, dDID, sdDID, dRIPV, sdRIPV, AT, Slope_a, S3, S4, NI, AI, AI1, RSD and RSC, are selected from the first 32 features with the lowest σ values, while the remaining two of the 32 selected features, which are S1 and S2, are not. The reasons why we include these two features are that their σs are relatively low and they are commonly used features that were reported to be highly related to BP in the literature. Most importantly, the reason why sdAS and sdDS, which are on the list of the first 32 features with the lowest σ values, are not selected is because that the performance of models whose input features contain sdAS and sdDS is worse than the performance of those whose inputs contain S1 and S2. Finally, a set of features (***η*^32^ × 2,176,188**), including **hr, AS, DS, AA, dAA, sdAA, DA, dDA, sdDA, PI, dPI, sdPI, dVI, sdVI, AID, dAID, sdAID, dDID, sdDID, dRIPV, sdRIPV, AT, Slope_a, S3, S4, NI, AI, AI1, RSD, RSC, S1** and **S2**, is selected as our final feature set. 

Figure 9 shows the distributions of the values of the first two features, dDA(a) and NI(b), and the last two features, sdAT(c) and t7(d), which have the lowest σ values. Figure 10 shows the distributions of the values of features S1(a), S2(b), sdAS(c) and sdDS(d).

### 4.3. Model of Deep Neural Network Predictor

Despite the fact that models built from LSTM units are the most frequently and widely applied for time series-related problems, in this study, we choose a different approach. We use a fully connected neural network, which is much simpler in terms of the number of parameters compared to an LSTM unit, to build up our core deep fully connected neural network, serving as a predictor of BP. We conduct tests on several models to determine the optimal number of hidden layers and neurons, following the approach in [30]. Finally, our model is introduced as follows.

Our model, as shown in Figure 3, is a six-layered structure of a fully connected neural network and the dimensions of the input layer are 1 × 32, which represents the features extracted from a single cardiac cycle. The numbers of hidden nodes in each layer are 2048, 4096, 8192 and 2048 and every node is fully connected to all nodes in the next layer. The activation function we use is ReLU, and the output layer has dimensions of 1 × 2, which are the estimated SBP and DBP, respectively. Before the training of our proposed deep neural network predictor, the selected feature set (***η*^32^ × 2,176,188**) is split into three parts, and each part contains 70%, 20% and 10% of the data, which serve as training, testing and validation datasets, respectively. As for the training process, a gradient descent optimizer, called “Nadam” [31], is applied to update all the variables, including w and b, in the model. In every epoch, the root mean square error (RMSE) and MAE are measured as loss functions for every 512 batches. For software implementation, we use Keras [32] to build the DNN model, “numpy” toolkits [33] for signal preprocessing and the “Heartpy” toolkit [24] for cardiac cycle segmentation.

### 4.4. Performance of Proposed Model

There are several mathematical methods and indices that are used to evaluate the validity of the regression model. Among them, the RMSE and MAE between the ground truth of BP and the estimated BP are the most widely used to gauge the performance of a BP estimation model. The definitions of the RMSE and MAE are shown in (10) and (11), respectively [34]:(10)$$\mathrm{RMSE}\;=\;\sqrt{\frac{\sum_{i=1}^{i=N}{\left(Z_{i}-Z_{i}^{’}\right)}^{2}}{N}},$$
and
(11)$$\mathrm{MAE}\;=\;\frac{\sum_{i=1}^{i=N}\left|\;Z_{i}-Z_{i}^{’}\;\right|}{N},$$
where *N* is the number of total BP samples (SBP or DBP) to be evaluated and Z and $Z^{\prime }$ are the ground truth BP (SBP or DBP) and estimated BP (SBP or DBP), respectively. The performance of our proposed model is assessed by the standards established by the Association for the Advancement of Medical Instrumentation (AAMI) [35] and the British Hypertension Society [4]. Additionally, two classical statistical approaches to evaluate a regression model, which are Bland–Altman analysis and Pearson’s correlation analysis, are conducted to evaluate our proposed model. Most importantly, at the end, the RMSE is computed to compare the performance of our work with others and the results will be further discussed and elaborated in the following sub-sections. Figure 11 shows the distribution of absolute error across 2,176,188 records of SBP and DBP.

#### 4.4.1. Performance Evaluation by AAMI Standards

An article by the Association for the Advancement of Medical Instrumentation (AAMI) [35] suggests that the average and standard deviation of error among numerous measurements of SBP and DBP should not be larger than 5 mmHg and 8 mmHg, respectively. Fortunately, our proposed deep neural network model fulfills the criteria suggested by the AAMI with averages and standard deviations equal to 3.21 mmHg and 3.35 mmHg for SBP and 2.23 mmHg and 2.44 mmHg for DBP across 2,176,188 records of SBP and DBP. 

#### 4.4.2. Performance Evaluation by BHS Standards

Table 2 shows the BHS standard for BP measuring devices and the performance of our model. From Table 2, the performances of our deep neural network estimator for both SBP and DBP satisfy grade A of the BHS standards, with 80.63% of error lower than 5 mmHg, 95.86% of error lower than 10 mmHg and 98.78% lower than 15 mmHg for SBP and 90.19% of error lower than 5 mmHg, 98.29% of error lower than 10 mmHg and 99.59% lower than 15 mmHg for DBP.

#### 4.4.3. Pearson’s Correlation and Bland–Altman Analysis 

Pearson’s correlation analysis is one of the most popular methods used to evaluate the validity of a regression model by computing Pearson’s correlation coefficient *r*, whose value ranges from -1 to 1 and it can be formulated as [36]:(12)$$r=\frac{\sum_{i=1}^{i=N}\left(x_{i}-\mu_{x}\right)\left(y_{i}-\mu_{y}\right)}{\sigma_{x}\sigma_{y}},$$
where N is the number of points ($x_{i}$, $y_{i}$) on the plot and μ and σ are means and standard deviations, respectively. The basic concept of *r* is to measure the degree of correlation between two signals, x and y. In our case, if our proposed model is perfect and error free, then the Pearson’s correlation coefficient of our model should be equal to 1. From Figure 11, the results show that r is equal to 0.977 between the ground truth of SBP and the estimated SBP and that r is equal to 0.947 between the ground truth of DBP and the estimated DBP across 2,176,188 records in our dataset, revealing the extremely high correlation between estimated BP and the ground truth of BP. Figure 12 shows the Pearson’s correlation analysis results for the estimation error for SBP and DBP.

A Bland–Altman plot [37] is used in analyzing the agreement of two different arrays. It is another way to test the difference between estimated BP and the ground truth of BP in our case. Figure 12 show the Bland–Altman analysis results for SBP and DBP. From Figure 12a, there are two horizontal lines, which are y = μ+1.96σ and y = μ−1.96σ, respectively, forming a range called limits of agreement (LOA), and 95% of data points are in the range. From Figure 12, the LOA for errors of SBP is (−9.38 8.76) mmHg. On the other hand, the LOA for errors of DBP is (−5.97 6.87) across 2,176,188 records, which confirms the accuracy of our proposed model. Figure 13 shows the Bland–Altman analysis results for the estimation error for SBP and DBP.

#### 4.4.4. Comparison with Other Works

To be honest, it is extremely difficult for us to do a fair comparison with prior works for the following reasons. First, most of the existing models used both ECG and PPG as inputs of the models. Second, even if the inputs of the models were PPG only, it is still difficult to compare them, since the datasets used in different studies may be different. Last, but not least, even if the model to be compared uses the same dataset as our model and also takes only PPG as input, the number of subjects (and cardiac cycles) used for training and testing, which will hugely affect the degree of generalization of a model, may still be different. However, despite all the impediments, it is still necessary to compare our proposed model to other works owing to the fact that through the comparison, we will be able to understand and judge our own model better. The comparison results are shown in Table 3.

From Table 3, it is clear that in terms of the accuracy and scale of the experiments, our proposed deep neural network model is one of the best so far and, hence, the generalized BP estimation model with PPG signals only has been achieved.

## 5. Conclusions and Future Works

In summary, in this study, we propose a fully connected DNN model to estimate SBP and DBP, from a PPG signal only. We perform feature selection based on big data analysis using 9000 subjects, 2,176,188 records of BP in total and 32 optimal features selected based on the proposed selection index. Finally, our proposed model reaches BHS Grade A and satisfies the AAMI standard. The MAE is as low as 3.21 mmHg and 2.23 mmHg for SBP and DBP, and the RMSE is as low as 4.63 mmHg and 3.21 mmHg for SBP and DBP, which outperforms all existing works using the same dataset (MIMIC II). 

For future works, we plan to apply other RNN-related models to see if the MAE and RMSE can be further decreased. A sequence to sequence model is also a promising direction for this topic. Phase changes in blood pressure can happen under various influences. Thus, further studies should try to involve the correlation between BP estimation and natural time analysis. Additionally, we would like to implement our model in wearable devices to help people at risk of high blood pressure to monitor their BP continuously in their daily activities. Hence, studies on data from people outside hospital should be conducted.

## Figures and Tables

**Figure 1 sensors-20-05668-f001:**
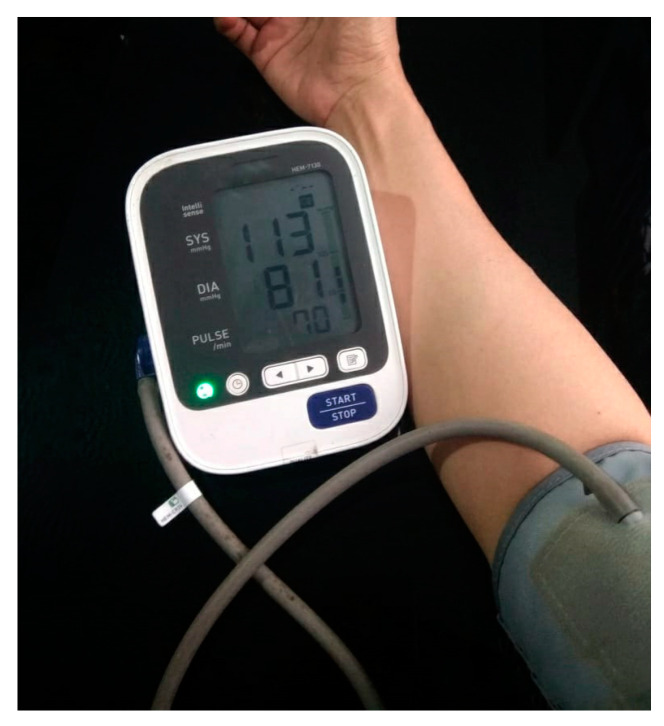
Traditional blood pressure measurement device.

**Figure 2 sensors-20-05668-f002:**
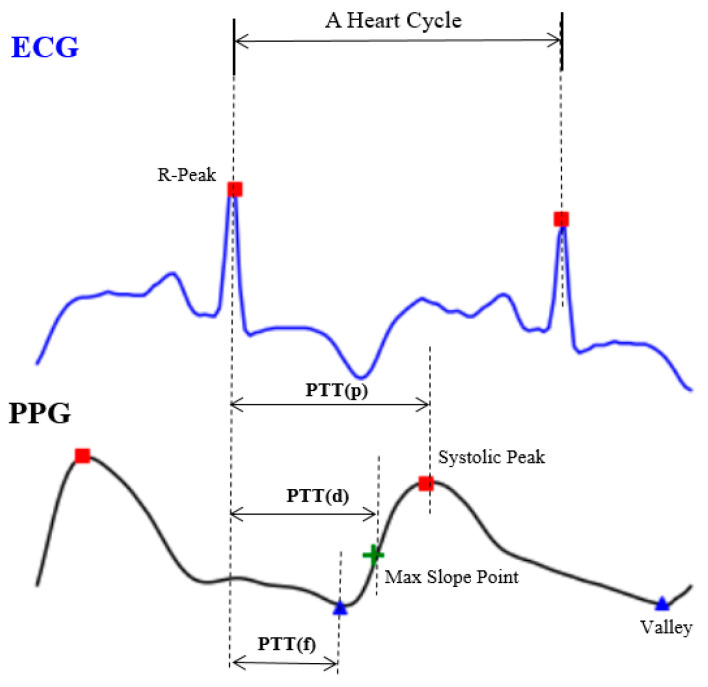
The expedient solution from engineers, including pulse transit time (PTT) (p), PTT (d) and PTT (f).

**Figure 3 sensors-20-05668-f003:**
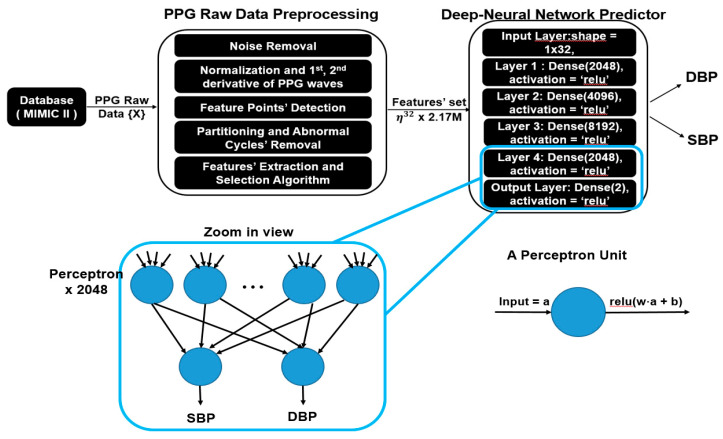
System overview of the proposed model and zoomed in view of fully connected neural network.

**Figure 4 sensors-20-05668-f004:**
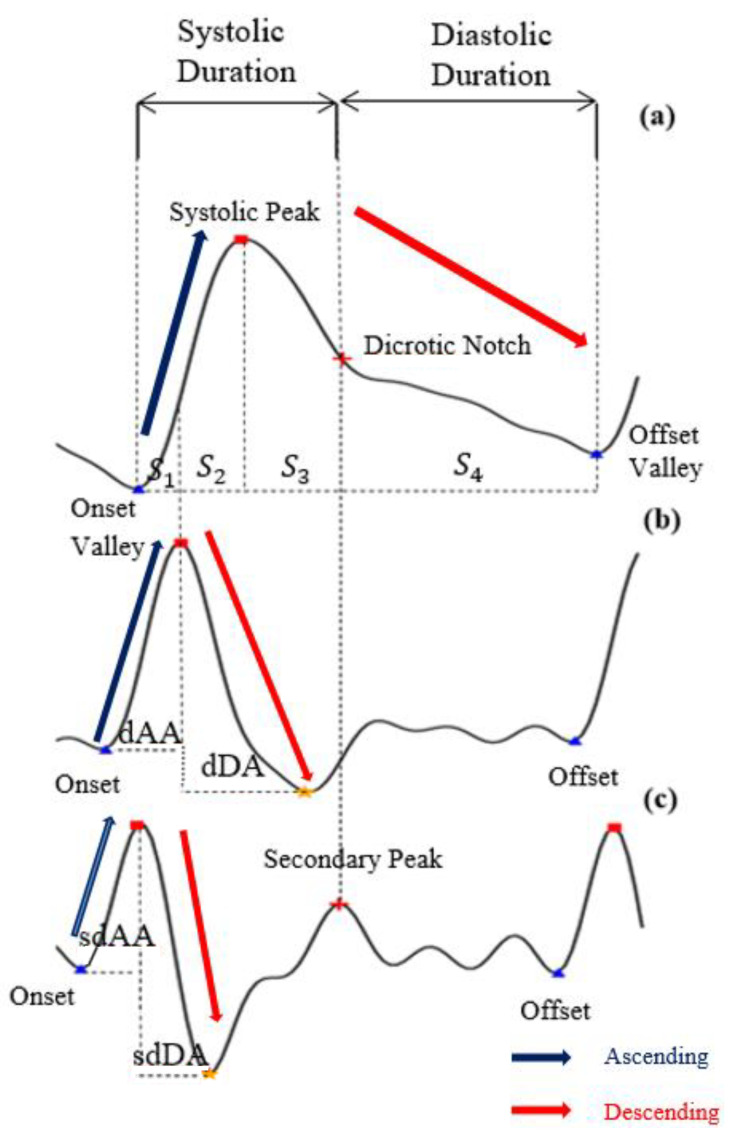
The results of feature point detection; (**a**) Photoplethysmogram (PPG), (**b**) 1st derivative of PPG (dPPG) and (**c**) 2nd derivative of PPG (sdPPG).

**Figure 5 sensors-20-05668-f005:**
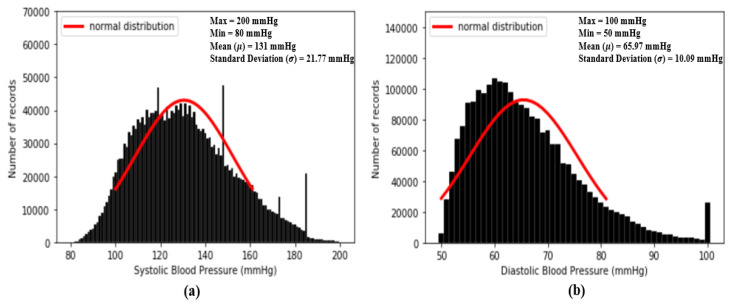
Histograms of distribution of blood pressure; (**a**) distribution of systolic blood pressure (SBP) (**b**) distribution of diastolic blood pressure (DBP).

**Figure 6 sensors-20-05668-f006:**
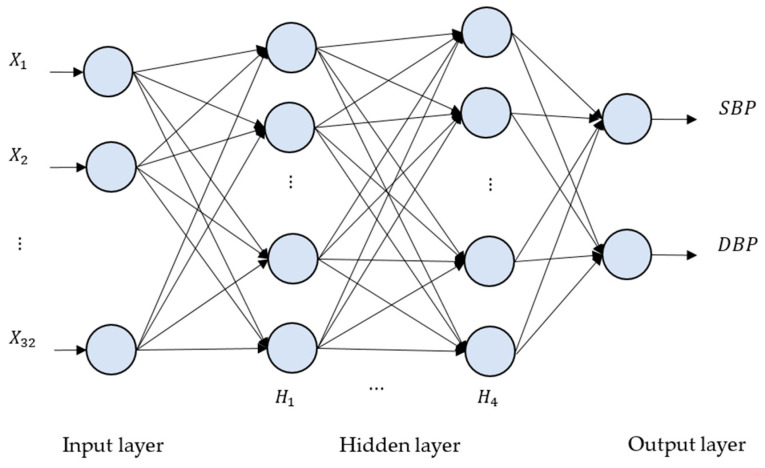
The deep neural network (DNN) architecture for the proposed method. There are four hidden layers, denoted as H1, …, H4. The numbers of neurons for H1, H2, H3 and H4 are 2048, 4096, 8192 and 2048, respectively.

**Figure 7 sensors-20-05668-f007:**
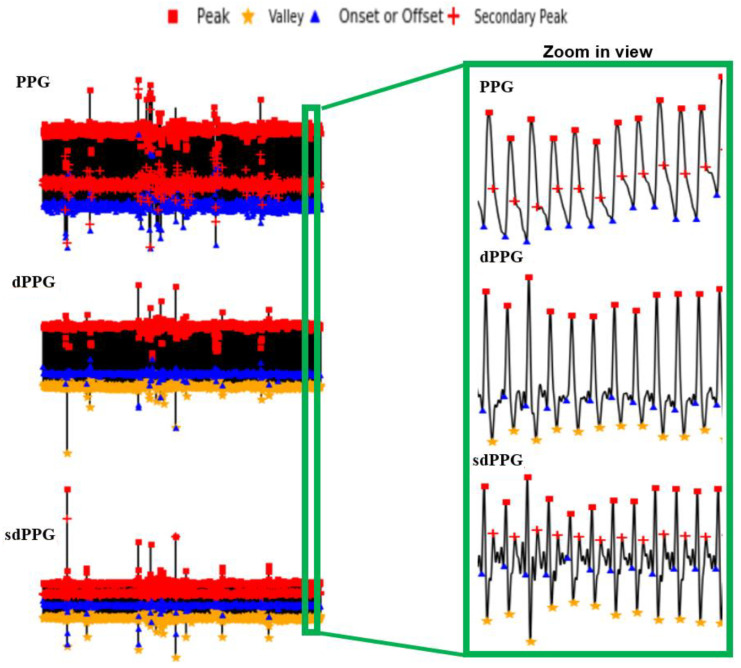
Examples of the results of feature point detection.

**Figure 8 sensors-20-05668-f008:**
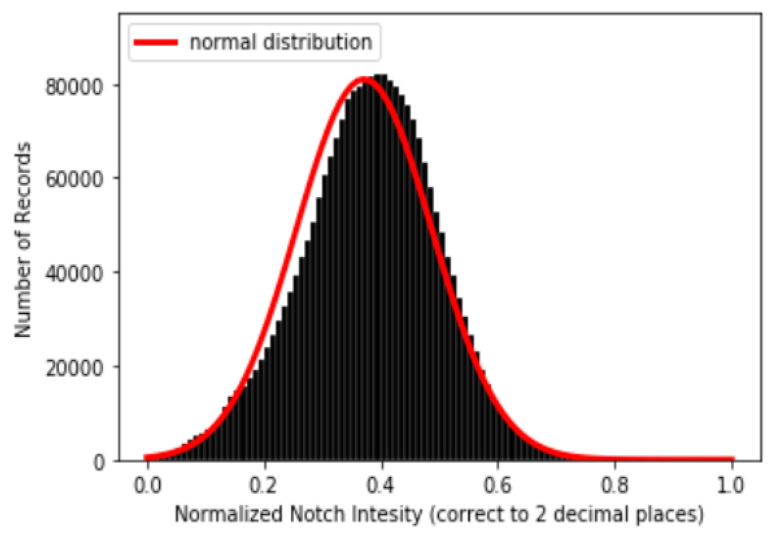
Distribution of normalized notch intensity.

**Figure 9 sensors-20-05668-f009:**
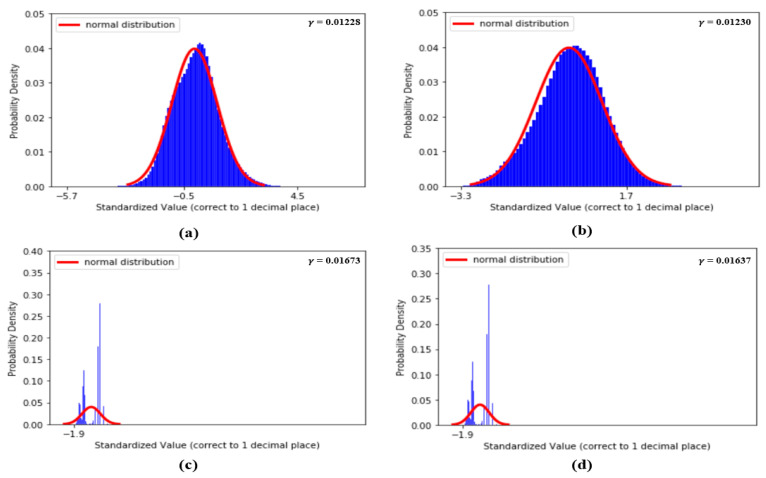
(**a**) Distribution of *dDA*, (**b**) distribution of *NI*, (**c**) distribution of *sdAT* and (**d**) distribution of *t7*.

**Figure 10 sensors-20-05668-f010:**
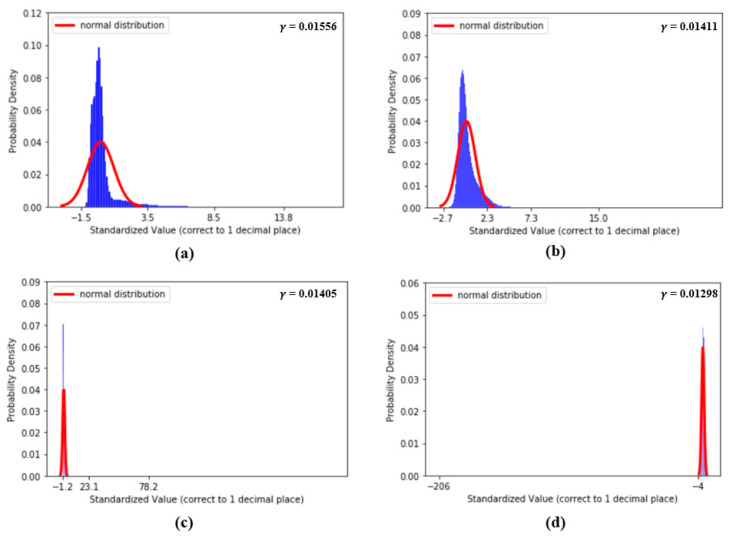
(**a**) Distribution of *S1*, (**b**) distribution of *S2*, (**c**) distribution of *sdAS* and (**d**) distribution of *sdDS*.

**Figure 11 sensors-20-05668-f011:**
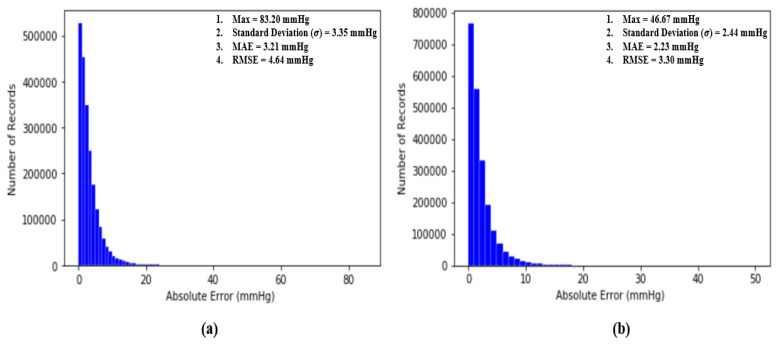
(**a**) Distribution of absolute error across 2,176,188 records of SBP and (**b**) distribution of absolute error across 2,176,188 records of DBP.

**Figure 12 sensors-20-05668-f012:**
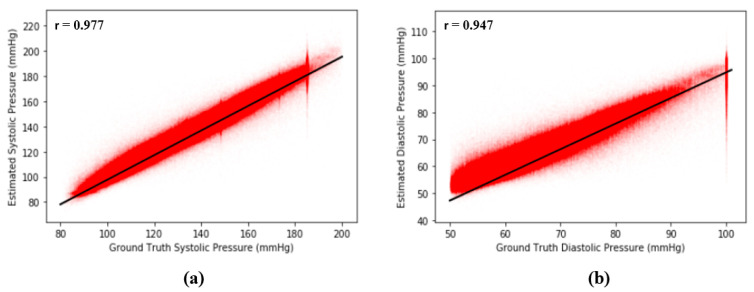
(**a**) Pearson’s correlation analysis results for error across 2,176,188 records of SBP and (**b**) Pearson’s correlation analysis results for error across 2,176,188 records of DBP.

**Figure 13 sensors-20-05668-f013:**
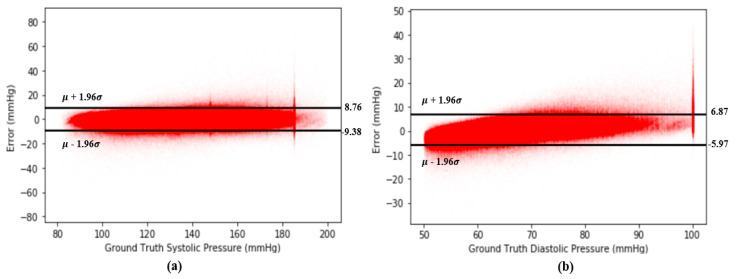
(**a**) Bland–Altman plot for error of SBP across 2,176,188 records and (**b**) Bland–Altman plot for error of DBP across 2,176,188 records.

**Table 1 sensors-20-05668-t001:** Part of the PPG feature definitions and corresponding γs are computed and listed in this table and all the definitions and denotations are in reference to past studies [10,16,25,26,27,28].

Denotation	Feature	Definition of Feature	σ
$\eta_{1}$	hr	Heart rate	^#^ 0.01281 ^*^
$\eta_{2}$	t1	Time interval of S1 (as seen in Figure 4)	-
$\eta_{3}$	t2	Time interval of S2 (as seen in Figure 4)	0.01653
$\eta_{4}$	t3	Time interval of S3 (as seen in Figure 4)	0.01637
$\eta_{5}$	t4	Time interval of S4 (as seen in Figure 4)	0.01604
$\eta_{6}$	t5	Time interval of dAA (as seen in Figure 4)	0.01634
$\eta_{7}$	t6	Time interval of sdDA (as seen in Figure 4)	0.01628
$\eta_{8}$	t7	Time interval of sdAA (as seen in Figure 4)	0.01673
$\eta_{9}$	t8	Time interval of sdDA (as seen in Figure 4)	0.01628
$\eta_{10}$	AS	Ascending slope of PPG (slope from onset point to maximum peak)	^#^ 0.01276 ^*^
$\eta_{11}$	dAS	Ascending slope of dPPG	0.01455
$\eta_{12}$	sdAS	Ascending slope of sdPPG	^#^ 0.01405
$\eta_{13}$	DS	Descending slope of PPG (slope from maximum peak to offset point)	^#^ 0.01405 ^*^
$\eta_{14}$	dDS	Descending slope of dPPG	0.01646
$\eta_{15}$	sdDS	Descending slope of sdPPG	^#^ 0.01298
$\eta_{16}$	S1	Area under PPG curve between onset point and maximum slope point (as seen in Figure 4)	0.01556 ^*^
$\eta_{17}$	S2	Area under PPG curve between maximum slope point and maximum peak (as seen in Figure 4)	0.01411 ^*^
$\eta_{18}$	AA	Ascending area of PPG (as seen in Figure 4)	^#^ 0.01381 ^*^
$\eta_{18}$	AA	Ascending area of PPG (as seen in Figure 4)	^#^ 0.01381 ^*^
$\eta_{19}$	dAA	Ascending area of dPPG (as seen in Figure 4)	^#^ 0.01255 ^*^
$\eta_{20}$	sdAA	Ascending area of sdPPG (as seen in Figure 4)	^#^ 0.01298 ^*^
$\eta_{21}$	DA	Descending area of PPG (as seen in Figure 4)	^#^ 0.01232 ^*^
$\eta_{22}$	dDA	Descending area of dPPG (as seen in Figure 4)	^#^ 0.01228 ^*^
$\eta_{23}$	sdDA	Descending area of sdPPG (as seen in Figure 4)	^#^ 0.01265 ^*^
$\eta_{24}$	RAAD	Ratio of ascending area to descending area, AA/DA	-
$\eta_{25}$	dRAAD	dAA/dDA	-
$\eta_{26}$	sdRAAD	sdAA/sdDA	-
$\eta_{27}$	PI	Peak intensity of PPG	^#^ 0.01261 ^*^
$\eta_{28}$	dPI	Peak intensity of dPPG	^#^ 0.01313 ^*^
$\eta_{29}$	sdPI	Peak intensity of sdPPG	^#^ 0.01305 ^*^
$\eta_{30}$	dVI	Valley intensity of dPPG	^#^ 0.01296 ^*^
$\eta_{31}$	sdVI	Valley intensity of sdPPG	^#^ 0.01299 ^*^
$\eta_{32}$	AID	Ascending intensity difference of PPG, intensity difference between maximum peak and onset point	^#^ 0.01324 ^*^
$\eta_{33}$	dAID	Ascending intensity difference of dPPG, intensity difference between maximum peak and onset point	^#^ 0.01311 ^*^
$\eta_{34}$	sdAID	Ascending intensity difference of sdPPG, intensity difference between maximum peak and onset point	^#^ 0.01305 ^*^
$\eta_{35}$	dDID	Descending intensity difference of dPPG, intensity difference between offset point and maximum peak	^#^ 0.01322 ^*^
$\eta_{36}$	sdDID	Descending intensity difference of sdPPG, intensity difference between offset point and maximum peak	^#^ 0.01310 ^*^
$\eta_{37}$	PIR	Peak intensity ratio of PPG, ratio of maximum peak intensity to onset intensity	-
$\eta_{38}$	dPIR	Peak intensity ratio of dPPG, ratio of maximum peak intensity to onset intensity	-
$\eta_{39}$	sdPIR	Peak intensity ratio of sdPPG, ratio of maximum peak intensity to onset intensity	-
$\eta_{40}$	dRIPV	Ratio of maximum peak intensity to valley intensity of dPPG	^#^ 0.01305 ^*^
$\eta_{41}$	sdRIPV	Ratio of maximum peak intensity to valley intensity of sdPPG	^#^ 0.01350 ^*^
$\eta_{42}$	AT	Ascending time interval of PPG	^#^ 0.01348 ^*^
$\eta_{43}$	dAT	Ascending time interval of sPPG	0.01634
$\eta_{44}$	sdAT	Ascending time interval of sdPPG	0.01673
$\eta_{45}$	DT	Descending time interval of PPG	0.01490
$\eta_{46}$	dDT	Descending time interval of dPPG	0.01628
$\eta_{47}$	sdDT	Descending time interval of sdPPG	0.01628
$\eta_{48}$	dTVO	Time interval between valley point and offset point of dPPG	0.01569
$\eta_{49}$	sdTVO	Time interval between valley point and offset point of sdPPG	0.01438
$\eta_{50}$	Slope_a	Slope from maximum peak to dicrotic notch of PPG	^#^ 0.01308 ^*^
$\eta_{51}$	S3	Area under PPG curve between maximum peak and dicrotic notch (as seen in Figure 4)	^#^ 0.01333 ^*^
$\eta_{52}$	S4	Area under PPG curve between dicrotic notch and offset point (as seen in Figure 4)	^#^ 0.01323 ^*^
$\eta_{53}$	RtArea	Ratio of systolic area to diastolic area, (S1 + S2 + S3)/S4 (as seen in Figure 4)	-
$\eta_{54}$	NI	Dicrotic notch intensity	^#^ 0.01230 ^*^
$\eta_{55}$	AI	Augmentation index = NI/PI	^#^ 0.01277 ^*^
$\eta_{56}$	AI1	Augmentation index 1 = (PI − NI)/PI	^#^ 0.01274 ^*^
$\eta_{57}$	RSD	Ratio of systolic duration to diastolic duration, (t1 + t2 + t3)/t4	^#^ 0.01405 ^*^
$\eta_{58}$	RSC	Ratio of diastolic duration to cardiac cycle, t4/(t1 + t2 + t3 +t4)	^#^ 0.01286 ^*^
$\eta_{59}$	RDC	Ratio of systolic duration to cardiac cycle, (t1 + t2 + t3)/(t1 + t2 + t3 + t4)	0.01611

“#” indicates that a value is one of the first 32 low γ values of the features. “*” indicates one of the selected 32 features. “-” suggests that the value is too large to be considered.

**Table 2 sensors-20-05668-t002:** The standards of the British Hypertension Society (BHS) for BP measuring devices and the performance of our model.

		Error ≤ 5 mmHg	Error ≤ 10 mmHg	Error ≤ 15 mmHg
BHS [4]	Grade A	60%	85%	95%
	Grade B	50%	75%	90%
	Grade C	40%	65%	85%
Our Model	SBP	80.63%	95.86%	98.78%
	DBP	90.19%	98.29%	99.59%

**Table 3 sensors-20-05668-t003:** Comparison of different models using PPG only as input for BP estimation.

Researchers	Dataset	Input	Performance
Mousavi et al. [38]	MIMIC II (441 subjects)	PPG	BHS standard: Grade A for DBP and Grade D for SBP AAMI: only the results of DBP satisfy the standards MAE, RMSE: not mentioned in the paper
Slapnivcar et al. [39]	MIMIC II (510 subjects)	PPG	MAE: DBP = 9.43 mmHg, SBP = 6.88 mmHg RMSE: not mentioned in the paper
Ibtehaz and Rahman [15]	MIMIC II (942 subjects)	PPG	BHS standard: Grade A for DBP and Grade B for SBP AAMI: the results of both DBP and SBP satisfy the standards MAE: DBP = 3.45 mmHg, SBP = 5.73 mmHg RMSE: not mentioned in the paper
Our proposed model	MIMIC II (**2,176,188 records of BP in total**)	PPG	BHS standard: **Grade A for both DBP and SBP** AAMI: the results of **both DBP and SBP satisfy** the standards MAE: DBP = **2.23 mmHg**, SBP = **3.21 mmHg** RMSE: DBP = **3.21 mmHg**, SBP = **4.63 mmHg**

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
