# Peer review of "Generalized Deep Neural Network Model for Cuffless Blood Pressure Estimation with Photoplethysmogram Signal Only"

_sensors, 2020, doi:10.3390/s20195668_

Round 1

Reviewer 1 Report

Dear authors,

the chosen topic of the work is relevant, but there are many doubts about the applicability of the obtained results to health care problems, which are described in the abstract. As is well known, the health care tasks described in the abstract require blood pressure recorders worn in daily activities. Because all patients lie in the Intensive Care Ward and the environment is completely different from their daily routine, the results obtained and validated based only on the Intensive Care (MIMIC) II online waveform database are unsuitable (until tested outside the hospital) for the tasks described in the abstract.

A lot of work has definitely been done, but  revisions must be done.

  • In the context of the presented study, the total number of heartbeats recorded during the whole study (22 row) is not informative: here the number of subjects and the number of studies performed for each subject are important.
  • (50-51 row) substantiate the statement why „is not a trivial task for wearable devices due to the coherence issue“
  • Verbal naming of the abbreviated PMV is required 1., 2 eq. In text (61row., 63row and ....)
  • 60 row: P – blood pressure, what is the pressure - systolic, diastolic, pulsed or other?
  • 60row is written: where P0 constanst, if so, then (eq1) is incorrect, because the subtraction of the constant P0 from the quantity P with units of measurement (dimension) does not make sense.
  • The statement 65-70 rows PTT explanation contradicts the basics of hemodynamics.
  • Between 55 – 70 rows, the reference to the same sources is repeated 4 times. This needs to be corrected.
  • 74 row, there is a reference to Figure2, but there is no Figure2 in the article.
  • 76 row, What does real-time BP monitoring mean? Does this mean beat-to-beat BP monitoring? If so, justify the relevance of real-time blood pressure measurement at home. If not, how long does it take to measure one systolic and diastolic blood pressure value?
  • 96 row, <Intensive Care (MIMIC) II online waveform database> used <for training, validating and testing for our deep neural networks predictor> Did you use all (MIMIC) II online waveform database records for training? Describe in detail the methodology used to compile the records for training, validating and testing.
  • Since the introduction deals with the application of the proposed method to wearable devices, it is likely that the target group is people outside the hospital. Therefore, the method must be tested in the target group.
  • In accordance with the BHS and AAMI standards, the blood pressure values ​​obtained must be compared with the values ​​obtained by the auscultatory method. Your work is not validated according to standards. Justify why this can be disregarded.
  • 101 row, What signal processing methods are used for „removing noise and baseline wander”.
  • The article contains textbook-type statements: „ 272 -74 rows The definition of RMSE and MAE is shown in Eq. (7) and (8), respectively“, „ 299 rows Pearson’s correlation analysis is one of the most popular methods“

Author Response

September 21, 2020

Response to Reviewer 1

Point 1: In the context of the presented study, the total number of heartbeats recorded during the whole study (22 row) is not informative: here the number of subjects and the number of studies performed for each subject are important.

Response:

Thank you so much for the suggestion. We have modified the sentence as follows (please refer to line 19-25 of page 1):

The effectiveness and accuracy of our proposed model was evaluated by the root-mean-squared error (RMSE), the Association for the Advancement of Medical Instrumentations (AAMI) standard and the British Hypertension Society (BHS) standard. Experimental results showed that the RMSEs in systolic blood pressure (SBP) and diastolic blood pressure (DBP) are 4.643 mmHg and 3.307 mmHg respectively across 9,000 subjects, with 80.63% of absolute errors among estimated SBP records lower than 5 mmHg and 90.19% of absolute errors among estimated DBP records lower than 5 mmHg.

Point 2: (50-51 row) substantiate the statement why „is not a trivial task for wearable devices due to the coherence issue“

Response:

Thank you so much for reviewer’s comment. We have modified our manuscript accordingly (please refer to line 46-49 of page 2).

Point 3: Verbal naming of the abbreviated PMV is required 1., 2 eq. In text (61row., 63row and ....)

Response:

Thank you so much for reviewer’s comment. We have modified our manuscript accordingly (please refer to line 60 and 62 of page 2).

Point 4: 60 row: P – blood pressure, what is the pressure - systolic, diastolic, pulsed or other?

Response:

Thank you so much for reviewer’s comment. This statement is rewritten in line 69 page 2. We meant blood pressure  is the blood pressure in unit of mmHg which can be the systolic, diastolic, or the pulse.

Point 5: 60row is written: where P0 constanst, if so, then (eq1) is incorrect, because the subtraction of the constant P0 from the quantity P with units of measurement (dimension) does not make sense.

Response:

Thank you so much for reviewer’s comment. We have modified the equation and the manuscript accordingly (please refer to line 57-70 of page 2).

Point 6: The statement 65-70 rows PTT explanation contradicts the basics of hemodynamics.

Response:

Thank you so much for reviewer’s comment. We have modified the explanation and the manuscript as follows (please refer to line 63-64 of page 2):

PTT is the time interval between where pulse wave is detected by two sensors and  is the distance between the sensors on the artery.

Point 7: Between 55 – 70 rows, the reference to the same sources is repeated 4 times. This needs to be corrected.

Response:

Thank you so much for reviewer’s suggestion. We have modified the citations and the manuscript accordingly (please refer to line 56-75 of page 2).

Point 8: 74 row, there is a reference to Figure2, but there is no Figure2 in the article.

Response:

Thank you so much for reviewer’s comment. We have put the Figure 2 in Line 78 of Page 3.

Point 9: 76 row, What does real-time BP monitoring mean? Does this mean beat-to-beat BP monitoring? If so, justify the relevance of real-time blood pressure measurement at home. If not, how long does it take to measure one systolic and diastolic blood pressure value?

Response:

Thank you so much for reviewer’s comment. Real-time BP monitoring means that the measurement can provide response/result within a specified timing constraint. We have mentioned at line 43-44 of page 1 that it can takes less than 10 seconds to do the measurements for both SBP and DBP value.

Point 10: 96 row, <Intensive Care (MIMIC) II online waveform database> used <for training, validating and testing for our deep neural networks predictor> Did you use all (MIMIC) II online waveform database records for training? Describe in detail the methodology used to compile the records for training, validating and testing.

Response:

Thank you so much for the suggestion. We have added the details in the Section 3.1 about the data source as follows: (Please refer to line 134~138 of page 4 and line 301-304 of page 12):

Based on the idea, Multi-parameter Intelligent Monitoring in Intensive Care (MIMIC) II online waveform database which has been refined and prescreened in the literature [8], containing 12,000 data instances, indicating unique subject’s records, and estimated 4M+ heart cycles, was used in this study. Among this enormous dataset, we preselect 9,000 data instances for training, validating, and testing for our deep neural network predictor.

Before training of our proposed deep-neural-network predictor, the selected features set (x 217,6188) are split into three parts, and each part contains 70%, 20%, 10% served as training, testing and validation dataset respectively.”

Point 11: Since the introduction deals with the application of the proposed method to wearable devices, it is likely that the target group is people outside the hospital. Therefore, the method must be tested in the target group.

Response:

Thank you so much for the suggestion. In order to perform fair comparison between our work and existing work, we have to use MIMIC II. Since the goal of this paper is to report the experimental results of our proposed method under academic setting, we do not have the environment or resources to enable us to conduct more experiments for patient outside the hospital. In the future, if we are funded to implement the proposed algorithm into the real wearable device which will be sold on the market, suggested studies on the target group will be conducted. We have mentioned this in the future works as follows (please refer to line 385~388 at page 17):

Also, we would like to implement our model on wearable devices to help people at risk of high blood pressure to monitor their BP continuously in their daily activities. Hence, studies on data from people outside the hospital should be conducted.  

Point 12: In accordance with the BHS and AAMI standards, the blood pressure values ​​obtained must be compared with the values ​​obtained by the auscultatory method. Your work is not validated according to standards. Justify why this can be disregarded.

Response:

Thank you so much for the comment. The AAMI standard and BHS standard are used for evaluating our results. The values are then being compared to the values obtained by machine learning and deep learning methods for fair comparison. (please refer to Table 4 at line 369 at page 15):

Table 4. Comparison upon different models using PPG only as input for BP estimation

Researchers

Dataset

Input

Performance

Mousavi et al. [38]

MIMIC II (441 subjects)

PPG

BHS Standard:

Grade A on DBP and Grade D on SBP

AAMI:

Only the results of DBP satisfy the standards

MAE, RMSE:

did not be mentioned in the paper

Slapnivcar et al. [39]

MIMIC II (510 subjects)

PPG

MAE:

DBP = 9.43 mmHg, SBP = 6.88 mmHg

RMSE:

did not be mentioned in the paper

Nabil Ibtehaz and Sohel Rahman [15]

MIMIC II (942 subjects)

PPG

BHS Standard:

Grade A on DBP and Grade B on SBP

AAMI:

The results of both DBP and SBP satisfy the standards

MAE:

DBP = 3.45 mmHg, SBP = 5.73 mmHg

RMSE:

did not be mentioned in the paper

Our proposed model

MIMIC II (2,176,188 records of BP in total)

PPG

BHS Standard:

Grade A on both DBP and SBP

AAMI:

The results of both DBP and SBP satisfy the standards

MAE:

DBP = 2.23 mmHg, SBP = 3.21 mmHg

RMSE:

DBP = 3.21 mmHg, SBP = 4.63 mmHg

Point 13: 101 row, What signal processing methods are used for „removing noise and baseline wander”.

Response:

Thank you so much for reviewer’s comment. We remove noise so does the baseline wander by turning off the low frequencies in range of 0 to 8 Hz. We have added the detailed explanation at line 140-145 of page 5.

Point 14: The article contains textbook-type statements: „ 272 -74 rows The definition of RMSE and MAE is shown in Eq. (7) and (8), respectively“, „ 299 rows Pearson’s correlation analysis is one of the most popular methods“

Response:

Thank you so much for reviewer’s comment. We have put the citation accordingly. Please refer to line 313 of page 12 and line 342 of page 14.

Reviewer 2 Report

Reviewer’s Recommendation

This manuscript needs minor revisions

Summary

An algorithm is presented (for implementation into wearable devices) for monitoring Blood Pressure (BP) alterations for the case of cardiovascular disease patients.

General comments

This manuscript presents an interesting technique for BP monitoring thus it exhibits many extracted findings from this research. Hard word has been done and this is recognized. Nevertheless, the authors should recheck their document as in some cases it is not so comprehensible. Also, a structure of this paper should be given in brief and in graphical form. This will help a lot for the diffusion of this work. Furthermore some works from literature are missing.

Suggested Improvements

Please see the following suggestions apart from the aforementioned and please import any applicable answers inside manuscript:

  1. You mention about the basic theory of classical method of extracting physiological parameters from ECG and PPG and thus significant methods should be mentioned but you have not studied works such as A Prototype Photoplethysmography Electronic Device that Distinguishes Congestive Heart Failure from Healthy Individuals by Applying Natural Time Analysis. Please revise.
  2. Figure 2 is not shown anywhere inside manuscript. Please revise.
  3. In Materials and Methods a general description of figure 3 should be given prior to mentioning the bullet-style composition of the model.
  4. You mention "where we smooth raw PPG data, removes the abnormal data following standard procedures suggested by [8] and partition data into 2.1 M clean heart cycles and extracts 32 features from each PPG heart cycle". Is is badly shaped. Please revise.
  5. In lines 101-103 you mention that you use FFT to remove frequencies. Please explain the method because as already known FFT (or IFFT as being a part of the linear pair FFT-IFFT) reverts time domain to frequency domain and vice-versa. You first analyze the spectrum and then you use probably a filter to cut the frequencies or not? This is extracted from your expression. Please revise.
  6. You mention that "each hidden layer contains 2048, 4096, 8192, 2048 fully connected neurons". Please explain why in the final layer there is the transition from 8192 to 2048 neurons.
  7. Figure 6 should correspond to your neural network and not being a general example. E.g. use a scale and ratios corresponding to your values.

Manuscript Rating:

  1. This manuscript needs minor revisions. Future work should definitely involve correlation to Natural Time Analysis.

Author Response

September 21, 2020

Response to Reviewer 2

Point 1: You mention about the basic theory of classical method of extracting physiological parameters from ECG and PPG and thus significant methods should be mentioned but you have not studied works such as A Prototype Photoplethysmography Electronic Device that Distinguishes Congestive Heart Failure from Healthy Individuals by Applying Natural Time Analysis. Please revise.

Response:

Thank you so much for reviewer’s suggestion. We have added a special section for literature review and mentioned the suggested work as follows (Please refer to line 94-98 of page 3):

“The clinical applicability of PPG is also verified in [17] which tried to distinguish individuals with congestive heart failure from the healthy one by applying the concept of natural time analysis (NTA). NTA is applied to analyze a phase change or critical point on a complex system such as human heart. The results obtained by PPG demonstrated a comparable value of accuracy towards the results obtained by ECG.”

Point 2:  Figure 2 is not shown anywhere inside manuscript. Please revise.

Response:

Thank you so much for reviewer’s comment. We have put the Figure 2 in Line 77 of Page 3.

Point 3:  In Materials and Methods a general description of figure 3 should be given prior to mentioning the bullet-style composition of the model.

Response:

Thank you so much for reviewer’s suggestion. We have modified the manuscript as follows (please refer to line 114 to 118 of page 4):

The main flow of the proposed model is illustrated in Figure 3, which composed of preprocessing, feature extractor, and deep-neural-network predictor. Since the raw PPG might contain noise and long sequence, preprocessing is necessary for further feature extraction process. In this section, the detailed explanation about each part is presented with given summary introduced as follows

Point 4:  You mention "where we smooth raw PPG data, removes the abnormal data following standard procedures suggested by [8] and partition data into 2.1 M clean heart cycles and extracts 32 features from each PPG heart cycle". Is is badly shaped. Please revise.

Response:

Thank you so much for reviewer’s suggestion. We have modified the manuscript accordingly. Please refer to line 119 to 127 of page 4.

Point 5:  In lines 101-103 you mention that you use FFT to remove frequencies. Please explain the method because as already known FFT (or IFFT as being a part of the linear pair FFT-IFFT) reverts time domain to frequency domain and vice-versa. You first analyze the spectrum and then you use probably a filter to cut the frequencies or not? This is extracted from your expression. Please revise.

Response:

Thank you so much for the suggestion. After we use FFT to convert the PPG signal into the frequency-domain, we remove noises so does the baseline wanders by turning off the low frequencies in range of 0 to 8 Hz. After that, we restore the signal into the time-domain by applying the IFFT. We have added the detailed explanation at line 140-145 of page 5.

Point 6:  You mention that "each hidden layer contains 2048, 4096, 8192, 2048 fully connected neurons". Please explain why in the final layer there is the transition from 8192 to 2048 neurons.

Response:

Thank you so much for reviewer’s comment. The selection for the number of neurons is basically based on several pilot tests we have conducted before. This approach is also conducted by previous study. We have added this explanation to the manuscript accordingly. Please refer to line 295-296 at page 12, as:

We conduct tests on several models to determine the optimal number of hidden layer and its neuron, following the approach in [30]. Finally, our model is introduced as follows.”

Point 7:  Figure 6 should correspond to your neural network and not being a general example. E.g. use a scale and ratios corresponding to your values.

Response:

Thank you so much for the suggestion. We have updated our figure accordingly. Please refer to line 226 of page 9.

Point 8:  Future work should definitely involve correlation to Natural Time Analysis.

Response:

Thank you so much for reviewer’s suggestion. We have added this suggestion into our conclusion and future works as follows (please refer to line 384-385 of page 16):

Phase change in blood pressure can be happen under various influence. Thus, further studies should try to involve the correlation between BP estimation and natural time analysis.

Round 2

Reviewer 1 Report

No comments.